# Behavioral Self-Blame in PTSD—Etiology, Risk Factors, and Proposed Interventions

**DOI:** 10.3390/ijerph20156530

**Published:** 2023-08-05

**Authors:** Aviad Raz, Ravit Rubinstein, Eran Shadach, Gal Chaikin, Ariel Ben Yehuda, Lucian Tatsa-Laur, Ron Kedem, Leah Shelef

**Affiliations:** 1Department of Health and Well-Being, IDF’s Medical Corps, Israel Defense Forces, Ramat Gan 6195001, Israel; 2School of Social Sciences, The Academic College of Tel Aviv-Yaffo, Tel Aviv Yaffo 6195001, Israel; 3Department of Military Medicine, Faculty of Medicine, The Hebrew University of Jerusalem, Jerusalem 91120, Israel; 4School of Social Work, Sapir Academic College, D. N. Hof Ashkelon, Sderot 79165, Israel

**Keywords:** PTSD, behavioral self-blame, feeling out of control, sense of control

## Abstract

Background: Feeling out of control during a traumatic event may evoke behavioral self-blame (BSB) to avoid feeling helpless following trauma by restoring one’s sense of control. BSB is a common, persistent, and treatment-resistant post-traumatic stress symptom. The present study investigates the etiology and risk factors of BSB following a traumatic event and the reasons for its persistence over time. Method: Subjects were a group of 546 Israeli ex-combat soldiers (M age = 24.93 ± 5.657) registered in an Israel Defense Forces (IDF) combat reaction clinic. All completed the Peritraumatic Dissociative Experiences Questionnaire (PDEQ), the Brief Symptom Inventory (BSI), and the PTSD Checklist for the DSM-5 (PCL-5). Item 10 of the PCL-5 served to measure BSB. The PDEQ and BSI measured distress and feeling out of control during the event. We used descriptive analyses of the data, *t*-test, and linear regression analysis to reveal the relationship between the research variables. Results: Feeling out of control during a traumatic event often increases BSB and post-traumatic stress symptoms. A significant correlation emerged between continuing distress characterizing individuals who experience a persistent lack of control and BSB. Female combat soldiers were at a higher risk of BSB than their male counterparts. Conclusion: Loss of control experienced during a traumatic event may result in persistent long-term feelings of lack of control over one’s behavior.

## 1. Introduction

Earlier exposure to severe injuries and fatalities predicts depression and anxiety symptoms [1]. Soldiers exposed to combat are at an elevated likelihood of clinical and severe functional impairment [2]. Moreover, combat exposes the participating soldiers to risks to their life, which may induce post-traumatic stress disorder (PTSD) [3]. Hence, battlefield experiences risk prompting the unique PTSD psychiatric syndrome [1,2,3].

The DSM-5 uses a four-factor model to characterize PTSD: Cluster B—intrusion symptoms, Cluster C—persistent avoidance of stimuli, Cluster D—negative alterations in cognitions and mood, and Cluster E—hyper-arousal and reactivity [4]. Cluster D covers, among other things, persistent distorted cognitions about the cause or consequences of the traumatic event(s), leading to self-blame or blaming others. Numerous studies have investigated the association between the guilt that often follows a traumatic event and PTSD [5,6,7]. Although theoretical and clinical models clearly implicate guilt in PTSD formation, its precise role remains uncertain given the existence of three differently measured guilt types—global guilt, characterological self-blame, and behavioral self-blame [8,9].

Most researchers have found a positive correlation between guilt and the intensity of post-traumatic stress symptoms (PTSSs). However, a consistent positive correlation emerged only with two guilt measures—negative cognitions about the self (a general negative view of the self) and negative cognitions about the world (blaming others). The findings about behavioral self-blame, a measure of one’s functioning during a traumatic event, were inconsistent. Although some studies suggested that behavioral self-blame contributes to developing PTSD [10], others argued that it may protect against it [11].

Janoff-Bulman [8] differentiates in her model between behavioral and characterological self-blame. She argues that behavioral self-blame is a control-related mechanism that involves attributions to a modifiable source (behavior during the traumatic event), whereas characterological self-blame is esteem-related and involves attributions to a relatively stable source (one’s character). The model hypothesizes that paradoxically, behavioral self-blame leads to weaker post-traumatic distress as it implies that the involved person has some control over the events, and is less likely to be affected by future traumatic experiences.

Feeling out of control is a common, well-known emotional reaction to traumatic experiences [4]. The definition of feeling out of control is broad, subjective, and context dependent. It generally refers to feeling incapable of managing emotions and behaviors. Feelings of being incapable of controlling one’s thoughts, emotions, and behavior frequently emerge in PTSD. When traumatized individuals feel out of control and unable to modulate their distress, they tend to develop unwanted or unfamiliar thoughts, emotions, and behavior (e.g., losing track of current events or feeling detached from the present reality).

Several studies have pointed out a link between feeling out of control during a traumatic event and guilt cognitions [12,13,14,15]. For example, loss of control during a traumatic event correlated positively with guilt cognitions in former prisoners of war [12]. Furthermore, both behavioral self-blame and characterological self-blame correlated positively with PTSD in people feeling out of control following a traumatic event. However, a more complex pattern emerged in people who maintained a sense of control under such circumstances: characterological self-blame continued to correlate positively with PTSD, whereas behavioral self-blame did not. Research results suggested that people who had experienced a loss of control during a traumatic event reacted with increased behavioral self-blame to avoid a sense of helplessness and regain control over the traumatic event, thus validating the suggestion that behavioral self-blame implies being in control [13]. The Conservation of Resources (COR) theory proposes a model to explain coping with stress and trauma. Its basic premise is that behind human actions stands a motivation to protect existing resources and acquire new ones [16]. Another central principle of the COR theory is that resource loss is more prominent than resource gain. Nonetheless, although resource gain may have little impact on people who do not experience specific loss or loss cycles, it is significantly more potent in cases of significant or ongoing resource loss [17]. Seen through the COR theory prism, although behavioral self-blame may cause mental pain, it is conducive to regaining a sense of control over a past event. By helping to advance toward rebuilding confidence, it becomes the preferable option.

### The Present Study

Despite the various research studies reviewed, the causes of behavioral self-blame persistence and resistance to treatment over time remain largely unknown. The results of a previous study [13] on PTSD in populations distressed by past events, such as military combat, serious accident, natural disaster, and sexual assault, revealed that feeling out of control during a traumatic event may evoke behavioral self-blame that helps individuals avoid feelings of helplessness following the trauma.

The purpose and the innovation of the present study was to investigate the etiology and risk factors (demographic and clinical) that cause behavioral self-blame following a traumatic event and the reasons why it tends to persist over time. These points have not been explored exhaustively in past research studies and promised to yield new insights.

Further to the previous findings of Raz et al. [13], our first hypothesis was that individuals who felt out of control during the traumatic event would demonstrate more behavioral self-blame and PTSSs than individuals who felt in control. Our second hypothesis was that long-term lack of control contributes to the persistence of BSB. To investigate the latter, we examined whether long-term distress (somatic, cognitive, emotional, and behavioral) leads to higher rates of behavioral self-blame, and therefore causes it to persist over time. We selected our research population from the existent comprehensive database of the IDF combat reaction clinic (see below). As this database does not include a built-in measure for lack of control, we leaned on the distress measure of the Brief Symptom Inventory (BSI), given the significant link found between trauma-related distress and a persistent sense of lack of control [18,19]. The findings of this research indicate that certain therapeutic interventions could prevent behavioral self-blame.

## 2. Materials and Methods

### 2.1. Design, Settings, and Procedure

The design of the current study is correlational. We used the IDF combat reaction clinic database to pick out our candidates for participation in the Shomer Eitan (i.e., “Steadfast Guardian”) project launched after the Protective Edge operation in the Gaza strip (July–August 2014). The operation involved 50 days of intense fighting, mostly urban warfare, and prolonged exposure to missile threats.

### 2.2. Participants

The participants were 546 IDF combat soldiers, aged 19 to 58 years (M = 24.93 ± 5.657), registered in the database of the IDF combat reaction clinic. The Shomer Eitan project, led by the Mental Health Department, intended to identify emergent anxiety syndromes following the Protective Edge Operation among compulsory-service and reserve soldiers via comprehensive psychiatric diagnostic analyses. All the candidates were interviewed by a psychiatrist or a clinical psychologist between October 2014 and July 2018, and completed a series of self-report questionnaires. Two of them did not complete the questionnaires, and the final number of cases investigated was therefore 544. The elapsed time since the mental (in some cases also physical) injury was 9.679 years (SD = 12.443).

### 2.3. Ethical Standards and Informed Consent

All procedures were in accordance with the ethical standards of the committee responsible for experimentation on humans (institutional and national) and with those of the 2000 revised edition of the Helsinki Declaration of 1975.

### 2.4. Measurements

#### 2.4.1. Demographic and Clinical Questionnaires

Demographic and clinical variables. Unless otherwise specified, details were valid for the Shomer Eitan project period. They included country of birth (Israel/other), gender (male/female), age, religion (Jewish/other), years of education (<12, 12, >12), socioeconomic status (low to average/high), type of military service (compulsory/career/reserve), rank (enlisted soldier/officer), adjustment difficulties on recruitment (Yes/No), motivation for combat service on recruitment (low/average/high), and contact with mental health services before Protective Edge Operation (Yes/No).

Socioeconomic status (SES) assessment. Categories were determined by residential area according to the Israeli Ministry of Interior 1–10 scale (10 being the highest socioeconomic status) derived from the Central Bureau of Statistics (CBS) classification system. The scale used was created by dividing the country geographically into residential clusters, i.e., by socioeconomic status, and numbering them from 1–10, with higher scores corresponding to higher socioeconomic status. Each soldier’s score was then converted to low-to-average (1–6) or high (7–10) status.

Military Adjustment Scale. The scale identifies traits that risk limiting functionality and adaptability to military service, and a mental health professional completes it at the recruitment center. It assesses the soldier’s ability to adjust to the military demands and indicates the military assignment that would be most suitable for a particular individual [20]. The definition of “adjustment difficulties” combines personality traits that are highly likely to limit functionality and the degree of adaptability to military service. The scale rates adjustment difficulties and has four levels: low (fit for combat duty); low-to-mild (fit for non-combat duty); moderate (fit for combat support); severe (special-needs soldier) [20]. However, in this study we only determined two categories—presence or absence of adjustment difficulties (Yes/No).

Motivation to serve in combat duties. A structured interview on recruitment assesses the motivation to serve as combat soldiers. The range of soldiers’ motivation scores is 8–40, divided into low (8–25), average (26–30), and high (31–40) [20].

#### 2.4.2. Dependent Variables

Behavioral self-blame. Since our database did not include a built-in measure for BSB, we used item 10 of the PCL-5 to assess perceived blame for the stressful experience or subsequent events. Although we acknowledge that this is not an optimal solution, using a single item for measurement is widely acceptable in medical outcome studies [21,22,23]. Notably, another item of the PCL-5 (item 20) emerged as strongly valid for measuring insomnia and was comparable to the complete Insomnia Severity Index [24].

Post-traumatic stress symptoms (PTSSs). To measure PTSSs, we used the PTSD checklist for the DSM-5 (PCL-5). The questionnaire includes 20 items, ranging from 0 (not at all) to 4 (extremely) on a Likert-type scale. The items corresponded to the four PTSD clusters of the DSM-5: intrusion (items 1–5); avoidance (items 6–7); emotional numbness (items 8–14); and hyper-arousal (items 15–20) [25]. In the present study, Cronbach’s alpha for PTSSs (item 10) was 0.93, both with and without behavioral self-blame.

#### 2.4.3. Independent Variables

Feeling out of control during the traumatic event. The Peritraumatic Dissociative Experiences Questionnaire—Self Report Version (PDEQ-10SRV) was designed to measure dissociation experiences during a traumatic event [26]. It assesses dissociative experiences that occur during and shortly after (minutes or hours) a traumatic event. The degree to which participants go through dissociative experiences is scored on a Likert scale (1 = not at all true; 2 = slightly true; 3 = somewhat true; 4 = very true; 5 = extremely true). The PDEQ-10SRV has been translated into Hebrew and validated [27]. The present study only uses items 1 (losing track of what was happening) and 5 (feeling detached from the moment’s reality) of the PDEQ-10SRV to differentiate out-of-control sensations from other dissociative experiences that may occur during a traumatic event. As we used only items 1 and 5, we established a cutting point to determine a decisive level of out-of-control sensation during a traumatic event in the following manner. An ROC curve analysis yielded a cutoff score of 1.75. Accordingly, participants who scored an average of 1.75 or higher in items 1 and 5 were included in the “feeling out of control” group; those whose average score was below 1.75 formed a “feeling in control” group. The previously continuous variable thus transformed into a categorical one. Cronbach’s alpha in the current study was 0.838.

Distress following the traumatic event. The Brief Symptom Inventory (BSI) is a self-administered questionnaire measuring distress and symptomatic behavior [28]. It consists of 53 self-report items rated on a five-point distress scale ranging from 0 (not at all), to 4 (extremely). The scale measures nine symptoms: depression, anxiety, hostility, interpersonal sensitivity, obsession-compulsion, paranoid ideation, phobic anxiety, psychoticism and somatization. Previous research has shown correlation coefficients of 0.90 to 0.92 between total and subscale BSI scores and SCL-90 scores [28]. The present study used the four BSI subscales representing emotional, behavioral, cognitive, and somatic distress.

Emotional Distress. The anxiety dimension subscale we used measures high-level anxiety symptoms (items 1, 12, 19, 38, 45, 49), nervousness, and tension, as well as panic attacks and feelings of terror, e.g., “Feeling so restless you can’t sit still”; “Feeling nervous or shaky inside.” Cronbach’s alpha was 0.78 [28], and in the present study it was 0.834.

Behavioral Distress. To measure this symptom, we used the hostility dimension subscale (items 6, 13, 40, 41, 46) evaluating characteristic anger-related actions and behaviors, such as a tendency to argue and outbursts, e.g., “Easily annoyed or irritated”; “Feeling an urge to beat-up someone.” Cronbach’s alpha was 0.72 and in the present study it was 0.828.

Cognitive Distress, measured by the obsessive-compulsive dimension subscale (items 5, 15, 26, 27, 32, 36) assessing thoughts of an unwanted nature experienced as unremitting and irresistible, e.g., “Trouble remembering things”; “Difficulty making decisions.” Cronbach’s alpha was 0.74 and in the present study it was 0.801.

Somatic Distress, measured by the somatic dimension subscale (items 2, 7, 23, 29, 30, 33, 37) that assesses distress arising from perceptions of bodily dysfunction, e.g., “It’s hard catching your breath”; “Faintness or dizziness.” Cronbach’s alpha was 0.75 and in the present study it was 0.825.

Total Distress scale, summing up the scores of the four scales. Cronbach’s alpha in the present study was 0.868.

### 2.5. Statistical Analyses

All the analyses used the Statistical Package for the Social Sciences (SPSS, version 20.0 for Windows). We set the level of statistical significance at *p* = 0.05. Analyses included:(a)Descriptive analyses of the data: mainly measures of central tendency, dispersion, and correlations.(b)Inferential statistics (null-hypothesis significance testing), accompanied by measures of effect size.(c)Mann–Whitney U test or Kruskal–Wallis test (non-parametric tests).(d)Pearson correlation between the study variables.(e)ROC curve analysis to determine the cutoff point between feeling in control vs. feeling out of control by identifying where “sensitivity” is equal to “specificity.” PTSD diagnosis served to assess the critical value. When the positive PTSD diagnosis and the cutoff point for feeling out of control were greater than or equal to 1.9, the sensitivity was 0.698 and the specificity was 0.361 (see Figure 1). As this value (around 1.9) did not appear in the present study’s data, the actual cut-point value was 1.75. The “feeling in control” group included subjects whose average scores for questions 1 and 5 in the PDEQ questionnaire were under 1.75, whereas the average score of the “feeling out of control” group was 1.75 or higher;(f)*t*-test comparing behavioral self-blame in the “feeling out of control” group vs. “feeling in control” group;(g)linear regression analysis to reveal the relationship between the research variables. The analysis used behavioral self-blame as the dependent variable; feeling out of control during the traumatic event and feelings of distress after the traumatic event (e.g., somatic, cognitive, emotional, and behavioral) were the independent variables. As mentioned in the Dependent Variables section, we used item 10 of the PCL-5 to assess perceived blame.
Figure 1Feeling of control during the traumatic event—ROC analysis.
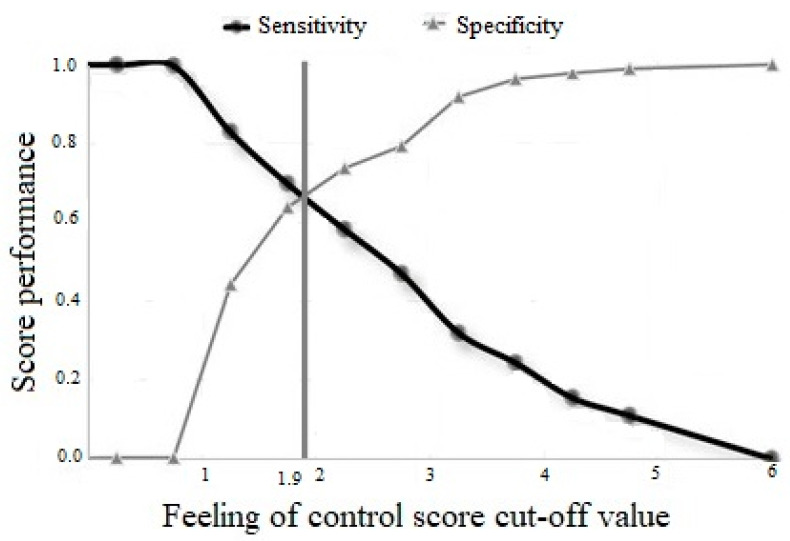



## 3. Results

The diagnostic criteria for probable PTSD were 33 points or more of the total PCL-5 score, and the appearance of all eight DSM-5 criteria for PTSD [29]. Of all the participants, 342 met the full DSM-5 criteria and 201 failed to meet one or more of them.

As explained above, we measured feeling out of control during the traumatic event as a categorical variable, and identified it via a receiver operating characteristic (ROC) analysis. Figure 1 shows the cut-point value of a positive PTSD diagnosis and feeling out of control during the traumatic event. The “feeling in control” group numbered 281 subjects, and the “feeling out of control” group numbered 242 subjects. A comparison of behavioral self-blame in the two groups revealed a significantly higher level of behavioral self-blame (t = 7.06, *p* < 0.001) with a strong effect measure of 0.62 in the “feeling out of control” group. A comparison of PTSSs in the two groups revealed a significantly higher level of PTSSs (t = 10.02, *p* < 0.0001) with a strong effect measure of 0.87 in the “feeling out of control” group. Note that effect measure (d) values of 0.20, 0.50, and 0.80 are considered weak, moderate, and strong, respectively, according to Cohen, 1988 [30].

Table 1 shows that female combat soldiers demonstrated more severe behavioral self-blame than male ones. Other demographic variables did not significantly affect the severity of behavioral self-blame.

Table 2 shows the correlations between behavioral self-blame, PTSSs, feeling out of control during the traumatic event, and distress following the traumatic event. Notably, all the correlations were positively significant.

Table 3 shows the results of the linear regression analysis. The analysis used behavioral self-blame as the dependent variable. The independent variables were feeling out of control during the traumatic event and feelings of distress following the traumatic event (e.g., somatic, cognitive, emotional, and behavioral). As seen in Table 3, only feeling out of control during the traumatic event and behavioral distress emerged as significant. Figure 2 presents the complete model.

## 4. Discussion

As hypothesized, behavioral self-blame and PTSSs were more prominent in soldiers who reported they felt out of control during a combat event than in those who felt in control. These findings match those of a previous study of 78 subjects aged 18 years and up, who went through different kinds of traumatic events such as military combat, serious accident, natural disaster, and sexual assault [13]. As mentioned, the COR theory maintains that individuals dealing with trauma and stress make every effort to regain the resources lost by the trauma. Life threats and fear highlight resource loss (i.e., loss of control and self-confidence). The COR theory also argues that behavioral self-blame may help trauma survivors regain control over a past event and proceed to rebuild confidence. However, behavioral self-blame risks becoming a source of severe mental pain that enhances PTSS (a positive correlation emerged between behavioral self-blame and PTSS). This finding is not surprising as the DSM-5 criteria for PTSD include self-blame [4]. By investigating the factors behind the long-term persistence of behavioral self-blame (The elapsed time since the injury was 9.679 years, SD = 12.443), we aimed to help trauma survivors cope with many years of behavioral self-blame after the traumatic event. To this end, we examined whether long-term distress (e.g., somatic, cognitive, emotional, and behavioral), known to be significantly related to a persistent sense of lack of control [18,19], led to higher rates of behavioral self-blame.

The results indicated that behavioral distress was the only factor leading to higher rates of behavioral self-blame, thus partially supporting our second hypothesis.

The theoretical implications. Behavioral self-blame may arise from an attempt to regain control over a past traumatic experience [13]. However, trauma survivors continue to feel that they are not in control for years after the traumatic event, and must cope with persistent uncontrollable behaviors causing them mental distress. We suggest that trauma survivors keep reverting to behavioral self-blame to cope with persistent feelings of lack of control. Another possible explanation for behavioral self-blame is that a sense of guilt over specific behaviors during the traumatic event may be unconsciously preferable to distress over various less-focused aspects of one’s ongoing behavior. Both explanations may help us understand why behavioral self-blame is remarkably treatment-resistant and persists over time.

There is general agreement that interventions are most effective immediately after the traumatic event but may also be beneficial after many years. This study indicates a need for different intervention approaches relative to the time elapsed since the traumatic event. Thus, shortly after the traumatic event, interventions should focus on enhancing one’s feeling of control over the experienced occurrences. For example, “YaHaLOM” (an acronym meaning Diamond), a mandatory program for IDF combat soldiers since 2017 [31], and the “iCOVER” curriculum for U.S. soldiers [32], are both designed to teach military personnel to identify acute stress reactions (ASRs) during combat events and respond appropriately to reduce PTSSs that may later develop into PTSD by controlling the situation during the traumatic event.

YaHaLOM and iCOVER are both prevention programs designed to be implemented immediately following the traumatic event and intended to reduce disorientation, regain cognitive control, and overcome helplessness sensations. We maintain that such interventions may also prevent future behavioral self-blame since individuals who feel in control during traumatic experiences are likely to demonstrate less behavioral self-blame than ones who feel they are out of control.

In addition to prevention interventions implemented shortly after the traumatic event, the present study’s results highlight the value of later therapeutic interventions in treating behavioral self-blame. The following treatment methods are the most recommended for soldiers diagnosed with PTSD: rapid eye movement treatment (EMDR), prolonged exposure (PE), cognitive processing (CPT), cognitive construction (CR), cognitive-behavioral treatments for trauma-focused therapy (TF-CBT), and stress management therapy (SMT) [33].

Practical implications. Our findings indicate that interventions to treat behavioral self-blame should focus on helping individuals reduce long-term behavioral distress that seems to play a crucial role in intensifying behavioral self-blame and causing it to persist over time. For example, Dialectical Behavior Therapy may improve emotional and behavioral regulation [34]. Cognitive-behavioral therapy (CBT) has emerged as the most effective and therefore the most commonly used approach to anger management [35]. Such interventions may also help reduce behavioral distress and treat behavioral self-blame.

Female combat soldiers emerged as being at increased risk of developing behavioral self-blame. Research studies suggest that females are more willing to concede responsibility for misdeeds and seem to have more difficulty disposing of feelings of guilt than males [36]. Furthermore, women are at increased risk for combat-induced mental health problems, including PTSD, sexual trauma, and suicide [37]. In a study of U.S. military personnel, women reported more distress due to combat experiences than men [38]. However, studies on females in combat situations are still scant, and the available data are insufficient to analyze in-depth gender differences [37,39]. The authors of this study speculate that female combat soldiers may be under more pressure to prove that they are up to the challenge, especially in combat situations, which may increase behavioral self-blame.

Another group that deserves further scrutiny is the officers. Officers are more likely to face the need to make life-or-death decisions and cope with a sharper dissonance between their perceived responsibility and control and combat situations that are often beyond control. Despite this, they do not display higher rates of behavioral self-blame. Officers are usually better informed about the military mission than regular soldiers and may consequently feel more in control. They bear responsibility for their subordinates’ lives. This responsibility, which has been correlated with a strong sense of control [40], may reduce their need for behavioral self-blame [13].

### Limitations

The main limitation of this study is that its correlational and retrospective design impairs the ability to conclude causality. Although this is typical of work with medical files, it should be borne in mind. A second limitation is that we had to use two items of the PDEQ as indicators of feeling out of control during a traumatic event. Although the authors are confident that this use yielded valid results, the PDEQ items were notably not designed to measure a sense of loss of control. A third limitation is that the variance presented in the study is biased, given the disparity in the time of the evaluation. Finally, behavioral self-blame was measured by a single item (item 10 of the PCL). Despite the mentioned limitations, the field-authenticity of the data used guarantees the results’ external validity. We believe that such studies are of great value and recommend further validation of our results in future research.

## 5. Conclusions

The present study suggests that feeling out of control during a traumatic event engenders a long-term lack of control, specifically in one’s behavior (anger-related actions and behaviors). Therefore, trauma survivors keep reverting to behavioral self-blame to cope with persistent feelings of lack of control, which may intensify PTSSs. We conclude, therefore, that an intervention performed immediately after the traumatic event should have a different focus than a later intervention. These findings warrant further research and validation in future studies.

## Figures and Tables

**Figure 2 ijerph-20-06530-f002:**
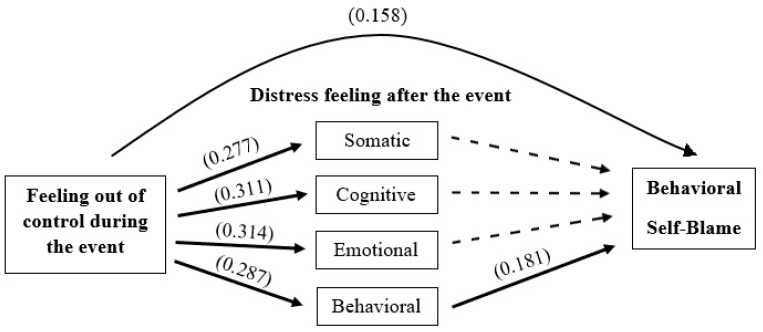
Model predicting Behavioral Self-Blame.

**Table 1 ijerph-20-06530-t001:** Demographic and clinical variables by behavioral self-blame. (**a**): Mann–Whitney U test (non-parametric test); (**b**): Kruskal–Wallis test by ranks (non-parametric test).

(**a**)					
		**Median**	**Interval Quarterly**	**U**	** *p* **
Gender	Female	3.00	3.00		
	Male	1.00	3.00	1884.5	0.035
Meetings with an MHO * prior to the traumatic event	No	1.00	3.00		
	Yes	1.00	2.00	18,870.00	0.255
Adjustment difficulties	No	1.00	3.00		
	Yes	1.00	2.50	5140	0.621
Country of birth	Israel	1.00	3.00		
	Other	1.00	0.00	9552	0.995
Religion	Jewish	1.00	2.00		
	Other	0.00	3.00	2774.5	0.181
Socioeconomic status	Low–Average	1.00	3.00		
	High	1.00	3.00	26,996.5	0.766
(**b**)					
		**Median**	**Interval Quarterly**	**H**	** *p* **
Type of service	Compulsory	1.00	3.00		
	Career	1.00	2.25		
	Reserve	2.50	−1.00	2.410 (df = 3)	0.492
Rank	Enlisted soldier	1.00	3.00		
	Officer	1.00	3.00	0.011 (df = 1)	0.916
	Low	1.00	3.00		
Motivation on the day of recruitment to serve in a combat unit	Average	1.00	−1.00		
	High	1.00	3.00	2577 (df = 2)	0.276
Education	<12	1.50	3.00		
	12	1.50	2.50		
	>12	0.00	3.00	1.208 (df = 2)	0.547

Note: Mann–Whitney *U* test (non-parametric test); * MHO’s-mental health officer. (**b**) Note: Kruskal–Wallis test by ranks (non-parametric test).

**Table 2 ijerph-20-06530-t002:** Correlations between Behavioral Self Blame, Post-traumatic stress symptoms (PTSSs), feeling out of control during the traumatic event (FOCD), and Distress Feeling after the traumatic event.

	Behavioral Self-Blame	PTSSs	FOCD	Somatic Distress	Cognitive Distress	Emotional Distress
PTSSs	0.562 **					
FOCD	0.292 **	0.4 **				
Somatic Distress	0.322 **	0.612 **	0.277 **			
Cognitive Distress	0.38 **	0.746 **	0.311 **	0.632 **		
Emotional Distress	0.382 **	0.788 **	0.314 **	0.667 **	0.732 **	
Behavioral Distress	0.403 **	0.651 **	0.287 **	0.469 **	0.593 **	0.643 **

Note: Pearson test; ** *p* < 0.001.

**Table 3 ijerph-20-06530-t003:** Linear regression for Behavioral Self-Blame as Dependent Variable.

Behavioral Self-Blame	B	Standardized Coefficient B	Sig *p*	95.0% Confidence Interval		Partial Correlation	Adjusted R^2^
				Lower	Upper		
Constant	0.073		0.574	−0.182	0.328		
FOCD	0.463	0.151	0.000	0.219	0.708	0.158	
Somatic Distress	0.014	0.060	0.259	−0.010	0.039	0.049	
Cognitive Distress	0.027	0.116	0.054	0.000	0.055	0.083	
Emotional Distress	0.015	0.066	0.310	−0.014	0.045	0.044	
Behavioral Distress	0.062	0.219	0.000	0.033	0.090	0.181	0.211

Note: Linear regressions; FOCD, feeling out of control during the traumatic event.

## Data Availability

Not applicable.

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
