# Peer review of "Behavioral Self-Blame in PTSD—Etiology, Risk Factors, and Proposed Interventions"

_ijerph, 2023, doi:10.3390/ijerph20156530_

Round 1

Reviewer 1 Report (Previous Reviewer 2)

I think it has been repaired satisfactorily.

Author Response

Thank you for your time and review report. 

This manuscript is a resubmission of an earlier submission. The following is a list of the peer review reports and author responses from that submission.

Round 1

Reviewer 1 Report

1. The first paragraph is only 1 sentence. Please revise it. 

2. The quality of Figure 2 is low. Please revise it. 

3. This paper will be even better if the authors try a new approach: structural equation modeling or path analysis. The model predicting behavior looks strange, especially since the direct, indirect, and total effects are not presented. 

4. What are the theoretical and practical implications of the study? 

5. The conclusion needs to be rewritten. 

Author Response

March 13, 2023

Dear Dr., Mark Szepesi

Assistant Editor, MDPI Romania,

Thank you for considering our manuscript, ijerph-2252107 for publication in IJERPH. We revised the manuscript in response to all the comments of reviewers #1. Below are our detailed responses.

Reviewer #1

Quality of English Language

( ) English very difficult to understand/incomprehensible
( ) Extensive editing of English language and style required
( ) Moderate English changes required
(x) English language and style are fine/minor spell check required
( ) I am not qualified to assess the quality of English in this paper

Comments and Suggestions for Authors

  1. The first paragraph is only 1 sentence. Please revise it. 

Reply: We revised the sentence accordingly, Lines 30-35.

  1. The quality of Figure 2 is low. Please revise it. 

Reply: We improved the quality of Figure 2 accordingly, Line 285.

  1. This paper will be even better if the authors try a new approach: structural equation modeling or path analysis. The model predicting behavior looks strange, especially since the direct, indirect, and total effects are not presented. 

Reply: We have thoroughly considered using SEM, but concluded that although the model appears to be suitable for SEM it does not, in fact, lend itself to SEM for the following reason: the mediating variable is not a latent one but a group of empirical variables we thought crucial to examine individually against the dependent variable. As the diagram shows, only one of the variables is significantly related to the stress variable after the traumatic event. Only if we had seen a clear connection of the other variables to the stress variable after an event, it would have been possible to create a common factor (a latent variable) that could be a mediating variable for all of the variables only if we had seen a clear connection between the other variables to the stress variable after an event, which was not the case. We thought it essential to illustrate that only one of the mediating variables is significantly related to the stress variable while the other variables are not.

  1. What are the theoretical and practical implications of the study? 

Reply: We emended the relevant text accordingly: theoretical implications –line 343; practical implications –line 376.

  1. The conclusion needs to be rewritten. 

Reply: We have rewritten the conclusion as suggested, lines 417-420

dix to the article.

Reviewer 2 Report

Please add 2-3 more papers from this topic that have been printed on this or a similar topic.

I think that the statistics are done correctly (there is no need to complicate things with the Mann-Kendel test).

Author Response

March 13, 2023

Dear Dr., Mark Szepesi

Assistant Editor, MDPI Romania,

Thank you for considering our manuscript, ijerph-2252107 for publication in IJERPH. We revised the manuscript in response to all the comments of reviewers #2. Below are our detailed responses.

Reviewer #2

Open Review

Quality of English Language

( ) English very difficult to understand/incomprehensible
( ) Extensive editing of English language and style required
( ) Moderate English changes required
( ) English language and style are fine/minor spell check required
(x) I am not qualified to assess the quality of English in this paper

Yes

Can be improved

Must be improved

Not applicable

Does the introduction provide sufficient background and include all relevant references?

(x)

( )

( )

( )

Are all the cited references relevant to the research?

( )

(x)

( )

( )

Is the research design appropriate?

( )

(x)

( )

( )

Are the methods adequately described?

(x)

( )

( )

( )

Are the results clearly presented?

(x)

( )

( )

( )

Are the conclusions supported by the results?

(x)

( )

( )

( )

Comments and Suggestions for Authors

Please add 2-3 more papers from this topic that have been printed on this or a similar topic. 

Reply: We added four papers to the reference list, line 70.

I think that the statistics are done correctly (there is no need to complicate things with the Mann-Kendel test).

Reviewer 3 Report

Comments to the Author

I read the present manuscript entitled “Behavioral Self-Blame in PTSD – Etiology, Risk Factors, and Proposed Interventions” with a major interest. In this study, the colleagues identified a gap related to the difficulty in identifying a pattern of this aspect of self-blame, especially in socio-emotional measures. The aspect of relevance is demonstrated in the investigation of the etiology and factors with risk probabilities that can lead to behavioral self-blame after a traumatic event.

The design of the study is elegant, the experimental procedure and statistical analysis offers an extensive description of the etiology and factors with risk probabilities. Combination of statistical tools help respond to these impacts of self-blame during post-traumatic stress events. The manuscript is clearly presented and highlights all the novelty of this work. I am in favor of publication.

Minor comment:

The objective of investigating the etiology and risk factors that cause behavioral self-blame after a traumatic event and its persistence over time could be more clearly directed towards the gap.

The statistics of the main findings could be present in the abstract.

What does the group think about stratifying the sample based on active duty vs. soldiers in reserve, for comparative purposes.

The variance presented in the study shows a bias that must be taken into account in the limitations of the study, the disparity in relation to the time for the evaluation.

It could make it clearer for the reader, which is the interpretation of the effect – if it is low, moderate or high. For research this is intuitive, however, from a clinical point of view this information is essential in order to measure comparisons.

Scatter plots for correlations, in order to have greater clarity in the distribution between variables, can be appreciable.

Author Response

March 13, 2023

Dear Dr., Mark Szepesi

Assistant Editor, MDPI Romania,

Thank you for considering our manuscript, ijerph-2252107 for publication in IJERPH. We revised the manuscript in response to all the comments of reviewers #3. Below are our detailed responses.

Reviewer #3

Quality of English Language

( ) English very difficult to understand/incomprehensible
( ) Extensive editing of English language and style required
( ) Moderate English changes required
( ) English language and style are fine/minor spell check required
(x) I am not qualified to assess the quality of English in this paper

Yes

Can be improved

Must be improved

Not applicable

Does the introduction provide sufficient background and include all relevant references?

(x)

( )

( )

( )

Are all the cited references relevant to the research?

(x)

( )

( )

( )

Is the research design appropriate?

(x)

( )

( )

( )

Are the methods adequately described?

(x)

( )

( )

( )

Are the results clearly presented?

(x)

( )

( )

( )

Are the conclusions supported by the results?

(x)

( )

( )

( )

Comments and Suggestions for Authors

Comments to the Author

I read the present manuscript entitled “Behavioral Self-Blame in PTSD – Etiology, Risk Factors, and Proposed Interventions” with a major interest. In this study, the colleagues identified a gap related to the difficulty in identifying a pattern of this aspect of self-blame, especially in socio-emotional measures. The aspect of relevance is demonstrated in the investigation of the etiology and factors with risk probabilities that can lead to behavioral self-blame after a traumatic event.

The design of the study is elegant, the experimental procedure and statistical analysis offers an extensive description of the etiology and factors with risk probabilities. Combination of statistical tools help respond to these impacts of self-blame during post-traumatic stress events. The manuscript is clearly presented and highlights all the novelty of this work. I am in favor of publication.

Reply: We thank the reviewer for his comment.

Minor comment:

The objective of investigating the etiology and risk factors that cause behavioral self-blame after a traumatic event and its persistence over time could be more clearly directed towards the gap.

Reply: We added an explanation accordingly, lines 351-352.

The statistics of the main findings could be present in the abstract.

Reply: We added the statistics of the main findings to the abstract, lines 19-21.

What does the group think about stratifying the sample based on active duty vs. soldiers in reserve, for comparative purposes.

Reply: A comparison between active duty vs. soldiers in reserve is presented in table 1.b. Kruskal Wallis test by ranks (non-parametric test) in three types of service: Compulsory; Career; Reserve. Line 292.

The variance presented in the study shows a bias that must be taken into account in the limitations of the study, the disparity in relation to the time for the evaluation.

Reply: We revised the limitations as suggested, Lines 407-411.

It could make it clearer for the reader, which is the interpretation of the effect – if it is low, moderate or high. For research this is intuitive, however, from a clinical point of view this information is essential in order to measure comparisons.

Reply: We added the interpretation of the effect, lines 261-265.

Scatter plots for correlations, in order to have greater clarity in the distribution between variables, can be appreciable.

Reply: Thanks for the comment. Indeed, the Scatter plots for correlations can contribute to understanding the whole picture. Yet, since there are 21 correlations in our study (as seen in Table 2), the analysis would have to include 21 graphs, too many for presentation. If needed, we could add them as an appendix to the article.

Round 2

Reviewer 1 Report

I believe that the paper is ready for publication.